# Shark Bite Reporting and The New York Times

**DOI:** 10.3390/biology11101438

**Published:** 2022-09-30

**Authors:** Christopher L. Pepin-Neff

**Affiliations:** School of Social and Political Sciences, Discipline of Government and International Relations, University of Sydney, Camperdown, NSW 2006, Australia; chris.pepin-neff@sydney.edu.au

**Keywords:** shark bite, shark incident, shark attack, media analysis, New York Times

## Abstract

**Simple Summary:**

This brief report looks at the language used by The New York Times to report on human-shark interactions and whether they go beyond the phrase “shark attack”. “Shark attack” is a phrase that holds a powerful psychological position in the mind of the public and directs the way sharks are talked about in society. However, it raises concerns because the phrase “shark attack” tells a one-dimensional story of shark behavior, which is often misleading. Data shows that between 32–38% of reported shark “attacks” have no injury at all. As a result, scrutiny of media reporting on different types of human-shark interactions is important. This study found The New York Times still uses shark “attack” language but has begun to layer it with more scientifically accurate shark “bite” and “sighting” narratives that convey less sensational stories. This is part of a growing trend to incorporate a broader lexicon that explains that not every interaction with sharks’ results in injury or fatality.

**Abstract:**

The social and political dynamics around human–shark interactions are a growing area of interest in marine social science. The question motivating this article asks to what extent media reporting by The New York Times has engaged beyond the lexicon of “shark attack” discourse to describe human–shark interactions. It is important because different styles of reporting on human–shark interactions can influence the public’s perceptions about sharks and support for shark conservation. This media outlet is also a paper of record whose editorial style choices may influence the broader media landscape. I review reporting language from The New York Times for 10 years between 2012 and 2021 (n = 36). I present three findings: first, I argue that The New York Times has had an increased frequency in use of the term “shark bite” to describe human–shark interactions. Secondly, I find that shark “attack” is still used consistently with other narratives. Third, there appears to be an increased use of “sightings; encounter; and incident” descriptors since 2020. The implication of this is a layered approach to reporting on human–shark interactions that diversifies away from a one-dimensional shark “attack” discourse.

## 1. Introduction

On 17 September 1865, The New York Times reported on their first human–shark interaction [1]. Peter Johnson was aboard the schooner *Catherine Wilcox* out of Lubec, Maine when he jumped into the water to retrieve a piece of wood. While in the water, he was “seized about the middle” by a fourteen-foot shark. The New York Times notes, “the case attracts attention because of the fact that the shark must have been of the species known as “maneater” which are common in the low latitudes but are rarely seen in the shoal water. The common shovel-nose shark of our water seldom if ever attack mankind”.

The focus of this study centers on the use of phrase “shark attack” and how this terminology can influence public perceptions. “Shark attack” represents one of the most emotive two-word phrases in the English vocabulary. This language utilizes a very specific wording to describe human–shark interactions in ways that have a profound impact on the public’s relationship with sharks [2]. Studies have shown that when people believe in intent-based narratives, such as shark “attack”, they are less likely to support shark conservation policies. Media reporting on human–shark interactions often includes the phrase “shark attack”, which suggests a very serious or fatal outcome. Pepin-Neff and Wynter state, “fear of sharks correlates with support for lethal policies, this association is powerfully mediated by perceptions of intentionality” [3], page 1. In addition, “shark attack” discourse has a powerful influence on fear toward sharks by telling a story about a creature that intentionally eats humans, rather than a fish which accidentally or mistakenly bites people on exceptionally rare occasions. Moreover, Ostrovki et al. (2021) add to this analysis by noting the importance of age in determining fear of sharks based on peoples’ exposure to difference media sources [4].

This historical interplay relative to emotions, words, and actual human–shark interactions motivates the research question. The research question asks to what extent media reporting by The New York Times engages beyond the lexicon of “shark attack” discourse to describe human–shark interactions. The aim of this study is to examine reporting styles over a decade and determine if there are any variations from the “shark attack” monopoly. The New York Times provides useful examinations for several reasons. First, it was founded in 1851 so it provides a degree of media continuity. Second, today it has 9 million subscribers [5]. This readership reinforces its position as a paper of record. In addition, The New York Times has a slow history of textual variation. The New York Times Manual of Style and Usage has been updated just four times since 1950. As a result, perceived adjustments and changes to human–shark interaction reporting are significant in the context of the media landscape.

The thesis of this study is that The New York Times has diversified around human–shark interaction narrative language. Indeed, I present three preliminary findings: first, I argue that The New York Times has had an increased frequency in use of the term “shark bite” to describe human–shark interactions. Secondly, I find that shark “attack” is still used consistently with other narratives. Third, there appears to be an increased use of “sightings; encounter; and incident” descriptors since 2020. The implications of these factors contribute to multiple narratives around human–shark interactions rather than a one-dimensional shark “attack” discourse.

This study is important because different styles of reporting on human–shark interactions can influence the public’s perceptions about sharks and support for shark conservation. This vocabulary is deployed as a media asset to strategically tap into the reader’s subconscious and motivate them to read or click on a news link. In a landmark study on the media and sharks Muter et al. (2012) found that “Shark attacks were reported at least 5 times more than conservation concerns or any other shark-related topic” [6]. This angle is commonly pushed by editors at newspapers and large media outlets in ways that give certain meaning (i.e., intentionality, fatal outcomes) which have survival value as click bait to events involving sharks. Moreover, there may be financial incentives to gain more clicks on highly sensationalized headlines. This can have implications for shark conservation because stoking fears of sharks through the media is easy to do. A study by Crossley found, “the general public grossly overestimates the number of non-fatal and fatal shark attacks, doubling the number of non-fatal and quadrupling the number of fatal shark attacks” [7]. Therefore, an examination of a leading media outlet may provide a snapshot into future trends regarding coverage and foreshadow public perceptions of sharks.

This study addresses several underlying tensions. First, the debate over the proper terminology to use when reporting on events involving sharks is difficult because there are real tragedies for individuals that occur and emotionally difficult periods for communities [2]. Several fatal incidents have occurred over the past decade, namely in Recife Brazil, Cape Cod, Reunion Island, and Western Australia where “attack” is appropriate because the motivation of the shark appears scientifically determinant [8,9,10,11]. Additionally, while this represents a fraction of events, it has both social and scientific purchase that is important to recognize.

Secondly, any new labels that push back against the cultural monopoly of “shark attack” discourse may be seen to be making light or watering down human tragedies. For instance, following a highly publicized article in the Guardian and The New York Times, comedian Stephen Colbert had a short segment on his show, “The Late Show” in July 2021 regarding attempts to relabel human–shark interactions, stating: “In order to change public perception of sharks, Australian scientists are seeking to rebrand shark attacks as interactions or incidents. As in, I’m sorry ma’am, a shark interacted with your husband’s torso and he’s experiencing a not being alive incident” [12].

In addition, Fox host Tucker Carlson [13] also ran a segment regarding this story under the picture, “Today in Liberal Lunacy”, stating: “In another sign of global progress, some government agencies in one country are no longer using the term shark attacks. They said it is a stigmatizing phrase that unfairly prejudices people against these subsurface carnivores”.

Thirdly, there are different styles of reporting on human–shark interactions that can influence the public’s perceptions about sharks and support for shark conservation. Scientific pushback against the “shark attack” media reporting began in earnest in 1916 in the United States and in the 1920’s in Australia, where shark “accident” was more commonly used by local governments [14]. In 1933, Australian surgeon Victor Coppleson wrote about “shark attacks” in the Australian Medical Journal, and this seemed to turn the tide on its more widespread adoption and use [15]. In 1958, a typology of shark attack categories was introduced following a New Orleans Shark Symposium. They stated [16], “In the tabulation that follows, we set up certain categories for the various kinds of shark attacks: Unprovoked Shark Attacks, Provoked Attacks, Boat Attacks and Air and Sea Disasters”. The Symposium’s work has transitioned to modern research analysis by the International Shark Attack File, which uses two main categories: ‘“Unprovoked bites” which are defined as incidents in which a bite on a live human occurs in the shark’s natural habitat with no human provocation of the shark.’ Additionally, ‘“Provoked bites” which occur when a human initiates interaction with a shark in some way” [17].

Fourthly a post-*Jaws* dimension to shark “attack” discourse began in the 2000’s. In 2000, during the “Summer of the Shark” in Florida there was a new pushback by shark scientists such as Robert Hueter, and more officially in 2012 and 2013. Washington Post journalist Juliet Eilperin, in her book Demon Fish [18], notably refers to human–shark interactions as shark “strikes”. Additionally, in 2013, Neff and Hueter proposed new categories for human–shark interactions because they found that in a 2009 New South Wales, Australia government report, approximately 38% of reported “shark attacks” had no injury [19].

The 2013 proposed categories by Neff and Hueter [15] include:

Shark sightings: Sightings of sharks in the water in proximity to people. No physical human–shark contact takes place.

Shark encounters: Human–shark interactions in which physical contact occurs between a shark and a person, or an inanimate object holding that person, and no injury takes place. For example, shark bites on surfboards, kayaks, and boats would be classified under this label. In some cases, this might include close calls; a shark physically “bumping” a swimmer without biting would be labeled a shark encounter, not a shark attack. This is the category most closely aligned to what happened to surfer Mick Fanning in 2015 [20].

Shark bites: Incidents where sharks bite people resulting in minor to moderate injuries. Small or large sharks might be involved, but typically, a single, nonfatal bite occurs. If more than one bite occurs, injuries might be serious. Under this category, the term “shark attack” should never be used unless the motivation and intent of the animal—such as predation or defense—are clearly established by qualified experts. Since that is rarely the case, these incidents should be treated as cases of shark “bites” rather than shark “attacks”.

Fatal shark bites: Human–shark conflicts in which serious injuries take place as a result of one or more bites on a person, causing a significant loss of blood and/or body tissue and a fatal outcome. Again, we strongly caution against using the term “shark attack” unless the motivation and intent of the shark are clearly established by experts, which is rarely the case. Until new scientific information appears that better explains the physical, chemical, and biological triggers leading sharks to bite humans, we recommend that the term “shark attack” be avoided by scientists, government officials, the media, and the public in almost all incidences of human–shark interaction.

Following the release of this research, Shiffman [21] noted, “The American Elasmobranch Society, the world’s largest professional organization of shark and ray scientists, has issued a resolution calling on the Associated Press Stylebook and the Reuters Style Guide to retire the phrase “shark attack” in favor of a more accurate (and less inflammatory) wording that is scaled to represent real risk and outcomes”.

The contested nature of human–shark interaction discourse can also be seen in international examples of the way institutes refer to shark incidents. In reaction to this growing trend, the Global Shark Attack File changed its name to the Global Shark Accident File (and uses both attack and incident) and the Australian Shark Attack File (founded in 1984) changed its name to the Australian Shark-Incident Database. They refer to themselves as “Australia’s leading source of shark bite data”. This is important to note because the Australian Shark-Incident Database is a joint partnership with Taronga Conservation Society Australia, Flinders University, and the Australian state of New South Wales’s Department of Primary Industries, a government portfolio. However, the International Shark Attack File in Florida still utilizes “shark attack” language in its title.

Cape Cod National Parks Service refers to shark incidents as shark bites [22]. The state of California has updated their language for reporting on human–shark interactions. The state of California’s Shark Incident Report [23] notes two things that are important. First, the use of the phrase “shark incident” is used and defined as: “* A shark incident is defined as any documented case where a shark approached and touched a person in the water, or touched a person’s surfboard, kayak, paddleboard, etc. This summary does not include shark sightings where no contact occurred, instances where sharks approached boats, or cases where hooked sharks caused injury or damage”.

According to the state of California statistics, 39% of shark incidents (79 of 203) dating from the 1950’s to the present involved no injury (see Figure 1). This is similar to the Neff and Hueter [14] analysis, which found that 38% of government reported human–shark interactions between 1979 and 2009 had no injury. In short, this is an important data point to receive affirmation by another source. Adding to this is data from the Australian Shark-Incident Database (Meagher, 2021) that shows 34% of incidents involve no injury (405 of 1196).

In addition, the Queensland Shark Management Plan 2021–2025 does not include the phrase “shark attack” but does use the phrase “shark bite” 13 times [24]. Additionally, the state of Western Australia’s 2021 report, entitled: “Results of the non-lethal SMART drumline trial in south-western Australia between 21 February 2019 and 20 February 2021” only uses the phrase “shark attack” when quoting others or within citations. They refer to “shark bite incidents” and shark “bite” 9 times.

The City of Cape Town uses “unprovoked shark bites” noting, “In South Africa, there have been 249 confirmed, unprovoked shark bites on humans in the past 111 years”. While the Cape Town funded Shark Spotters program uses both “shark attack” and “shark bite incident” language on its website, stating “The last shark bite in Cape Town was in August 2014 just off Sunrise beach near Muizenberg”.

Fifth, this contestation of language is consistent with public attitudes. In two different surveys, respondents indicated that “shark attack” is overused [3]. The studies conducted of residents in Australia in Ballina (n = 500) in 2015 and Perth (n = 600) in 2016 following a series of shark bites affecting those communities, respondents were asked in the two separate surveys if they “agree that the term “shark attack” was too sensationalized by the media. Data show that 66% of respondents from Ballina (n = 331) and 67% from Perth (n = 400) agreed that the term shark attack was too sensational, while in both 32% disagreed, with Ballina (n = 161) and Perth (n = 190). In both cases, just 2% were unsure. This survey involved 400 people in the Federal Seat of Perth as well as an over sample to ensure 100 from each of the two locations in Mundurah and Mindarie. In addition, 53% of Ballina residents stated shark bites are “accidental”, with just 22 percent describing them as “intentional” and the remaining 25 percent undecided. In the Perth survey, a majority (52%) of those surveyed stated that they believed shark bites are accidents, rather than intentional (22%) and not sure (26%). Therefore, the underpinnings of the “shark attack” narrative appears to be dissipating. Additionally, a larger majority (59%) of those surveyed in Perth also believed that “no one” was to blame when shark bites occur”.

Moving forward, this report reviews reporting language for 36 articles from The New York Times between 2012 and 2021 with the lexicon of human–shark interactions in mind to conduct a media content analysis of reporting on human–shark interactions. For instance, a snippet of text, 295 words from The New York Times in 2020 shows the word “attack” appear 13 times, or once every 23 words (see Figure 2).

## 2. Materials and Methods

The material and methods are reviewed by looking at the reasons for selecting the case study of The New York Times, the gathering of data, and operationalize the data.

The New York Times media reports were chosen as the unit of study because it is the recognized paper of record in the United States, and a leading publication of influence around the world. In addition, they are an authoritative publisher of stories replated to human–shark interactions, having published their first story in 1865, covering the 1916 shark disaster in New Jersey, and the summer of the shark in 2000.

New York Times articles were selected following a Factiva search that was refined based on the publication, publication date, and keywords. A total of 36 New York Times newspaper articles were selected from between 2012 and 2021 to provide a suitable period from the introduction of Neff and Hueter [14] to the present. Articles were gathered chronologically using Factiva software with the primary keyword being “shark attack” (n = 14). Secondary keywords were gathered from 36 articles and included “shark bite” (n = 12), shark “sighting” (n = 7) “incident” (n = 5), and shark “encounter” (n = 3). Keywords were selected consistent with other media analyses of human–shark interactions and were inclusive of headlines and article body text. Articles were filtered to ensure they are addressing human–shark interactions rather than loan sharks, Baby shark, card sharks, or the names of sports teams. Daily television listings with recaps were also removed. The number of incidents per 5-year average is 72, with 5–6 fatalities [17]. The average number of reported bites is greater than the number of New York Times articles on human–shark interactions each year; however, it is noteworthy that 2020 recorded the highest number of fatalities since 2013, see Table 1.

Data were looked at in context by printing out and reading each article. Headlines were noted as well as the article text. The discourse and lexicon of human–shark interactions includes: “incidents, shark bite, shark attack, shark sighting, and shark encounter”. Articles were examined for these key words. Proper nouns were included, such as the International Shark Attack File because they reinforce (perhaps more so than any quote or comment) the tolerance for the specific language. Random articles with the word “shark attack” were removed to establish the 10-year core articles. Results were double-checked to ensure inter-coder reliability by a research student.

## 3. Results

In a review of 36 New York Times articles on human shark interactions across 10 years (2012–2021) there were several findings. Data (descriptor mentions) were analyzed and separated for content analysis based on the way they highlighted different discourse usage.

(1)“Shark attack” or “attack” appeared in the headline in 11 of 36 articles (30%). One article (Barnard, 2021) had “shark sighting” in the headline [14]. No articles had shark “bite” in the headline. Each article examined was written by a different author.(2)In Figure 3, shark “attack” and shark “bite” mentions are compared and highlight a potential shift in reporting style at The New York Times that provides new emphasis on shark “bite” as a descriptor for human-shark interactions since 2018.(3)In Figure 4, all keywords (bite, attack, sighting, incident, encounter) were compared by year. The data show that all The New York Times *articles* on human–shark interactions that mentioned shark “attacks” in 2021 (n = 5) and 2020 (n = 3) also mentioned shark “bites”. This was also the case in 2018 (n = 3).(4)There were nearly three times the number of mentions of shark “bite” in 2021 (n = 34) as there were in 2018 (n = 13), see Figure 5. This data also suggests an increase in “shark bite” descriptor usage at The New York Times since 2018.(5)Shark “sighting” is the third most common phrase to be mentioned in New York Times articles in 2021 (n = 7 mentions) and 2020 (n = 4 mentions).(6)Overall, while shark “bite” mentions have increased, the data highlight that individual mentions of shark “attack” per year at The New York Times still constitute the most prevalent language selection in reporting on human-shark interactions in 2021 (n = 51). Indeed, two articles in 2021 in particular account for some of the rise in shark “bite” descriptors with (n = 12 mentions) and (n = 16 mentions), respectively.

**Figure 2 biology-11-01438-f002:**
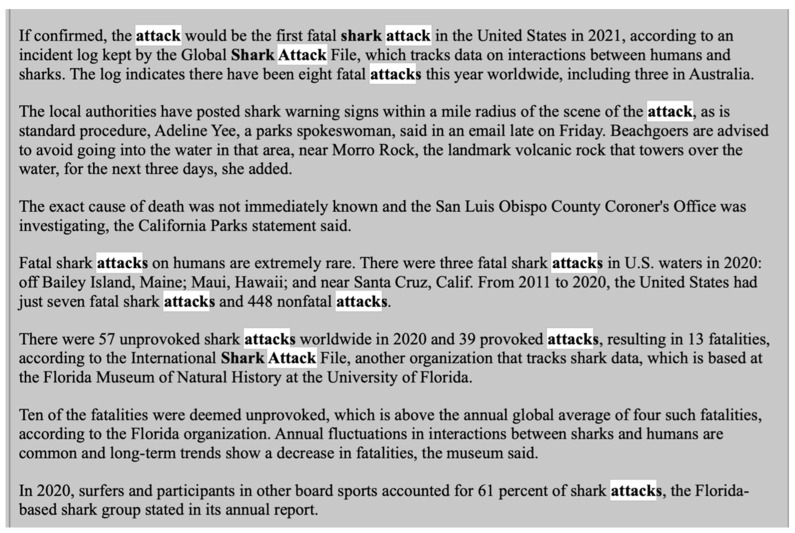
New York Times article with “attack” highlighted in the text.

**Figure 3 biology-11-01438-f003:**
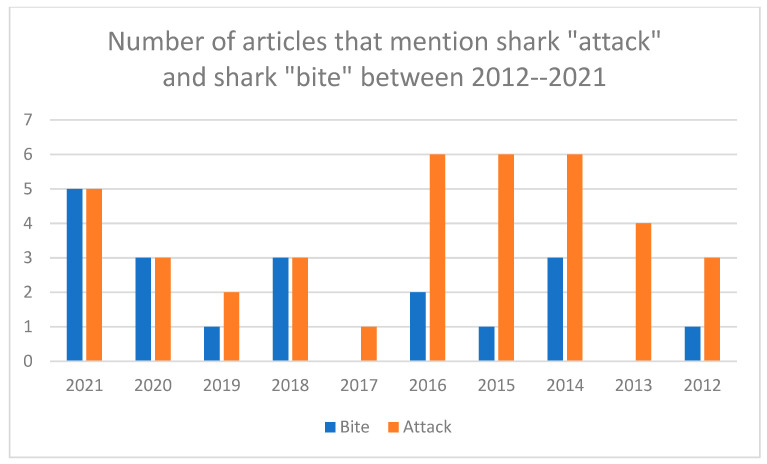
Contains data on the number of articles per year that mention each descriptor for human–shark interactions (n = 36).

**Figure 4 biology-11-01438-f004:**
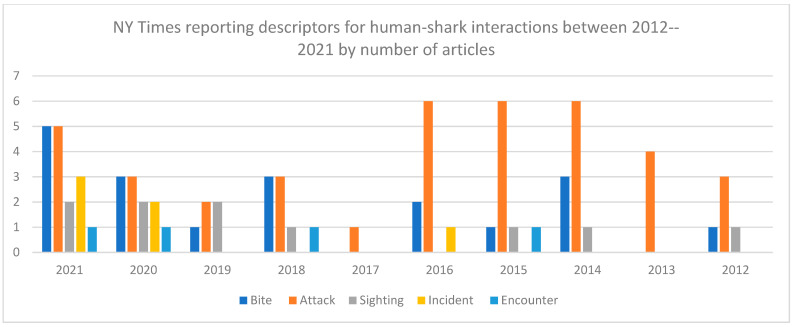
Number of descriptors mentioned per year in The New York Times articles (n = 36).

**Figure 5 biology-11-01438-f005:**
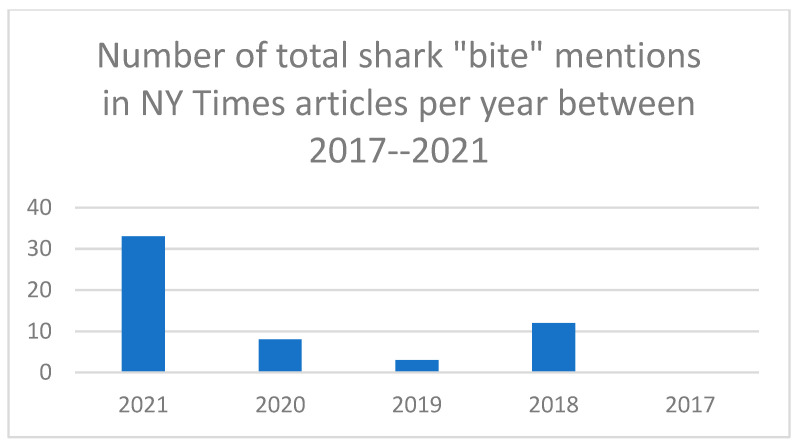
Contains data on the number of total shark “bite” mentions in NY Times articles per year between 2017–2021.

## 4. Discussion

The preliminary results suggest two types of responses over the 10-year period between 2012 and 2021. The first is based on the nature of individual occurrences and may reflect an individual reporter’s discursive preferences (between 2012–2017). The second; however, points an institutional response that is more layered and suggests a change in reporting style at The New York Times regarding human–shark interactions (since 2018).

The data indicates an increased use of varied discourse in recent years including shark “bites, attack, incident, sighting and encounter”. These terms represent a new approach by the media toward a layered reporting on human–shark interaction. This data can be interpreted from the perspective of previous studies on media relations and sharks [12,13,14,15]. I find that The New York Times is diversifying their discourse around human–shark interactions and that this is consistent with a growing international trend away from the singular adoption of “shark attack” as an applied phrase. The implications of this research highlight the importance of a varied lexicon and vocabulary to represent the multi-dimensional nature of human–shark interactions. Future research should examine broader trends in media reporting, particularly as scientific bodies focus on the Associated Press’ Style Guide as another media industry standard.

The adoption of a layering approach to media reporting on human–shark interactions would mark the end of the cultural monopoly of “shark attack” language utilization as a catch-all term. Shark “attack” is a one-dimensional representation that assumes knowledge of shark behavioral intent, sensationalizes human–shark interactions, provides a blanket label that assumes all sharks are dangerous, can be used politically to undermine shark conservation [25] and fails to scientifically represent the variety of possible experiences that people can have with sharks. The implications of this specific phrase’s use include harm to human populations because the nature of the risk posed by sharks is misrepresented and leads to a fundamental misunderstanding of sharks. If the public knew that nearly 40% of reported shark “attacks” had no injury this could change a building block of marine science education [18]. More varied text in media portrayals illustrates the variety of different types of human–shark interactions that are important to public communication and risk mitigation.

In all, human–shark interaction research is a growing area of marine social science. This article illustrates just one aspect of how human–shark interactions has become a growing interdisciplinary sub-field of human–shark relations within marine social science [26]. This emerging sub-field looking at human–shark relations relative to marine biology [27], geography [28], political science [29], conservation, film studies [30], communications [31]. This article contributes to a demonstrated “shark turn” in social science research. Further research is needed, but this study suggests that that there is evidence to support an emerging trend in diversified descriptions regarding the collection and reporting on human–shark interactions at a key media outlet.

## 5. Conclusions

The question driving this brief report asked to what extent media reporting by The New York Times engages beyond the lexicon of “shark attack” discourse to describe human–shark interactions. I found that there has been an increase in diversified language used in reports about human–shark interactions at The New York Times, including shark bites, attack, encounters, sightings, and incidents since 2020.

Additionally, “shark attack” discourse has a powerful influence on fear toward sharks by telling a story about a creature that intentionally eats humans, rather than a fish that accidentally or mistakenly bites people on exceptionally rare occasions. Finally, it is useful to note that the Neff and Hueter (2013) typology is applicable more broadly and could be used for other animal encounters, both water-based and terrestrial.

## Figures and Tables

**Figure 1 biology-11-01438-f001:**
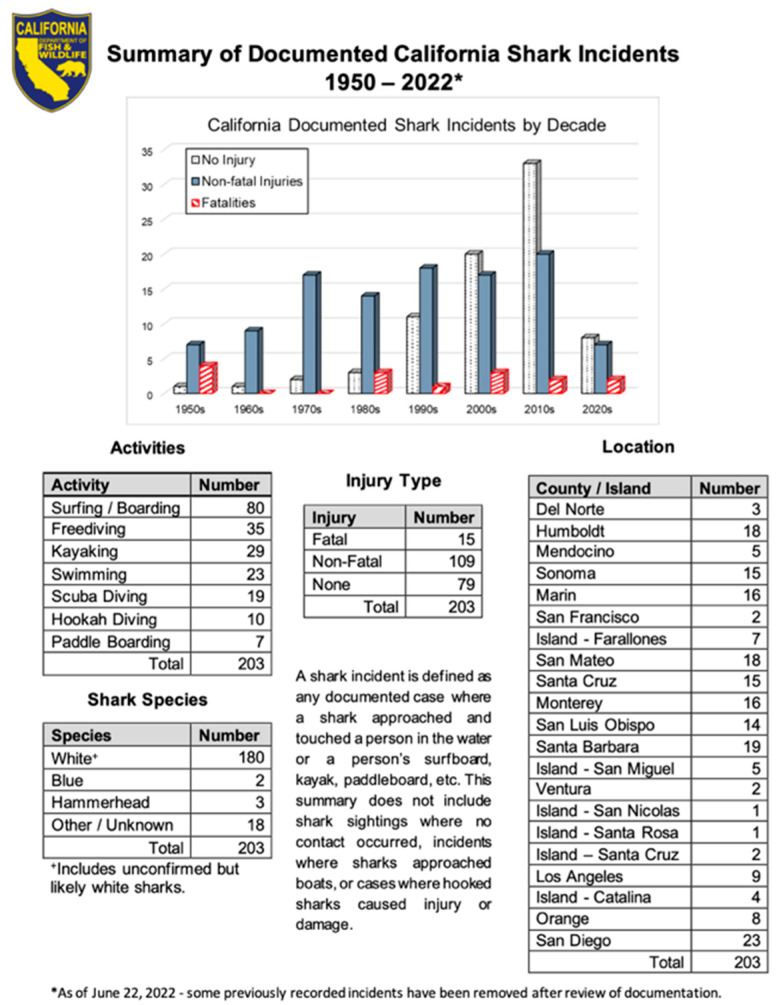
State of California Report on Shark Incidents.

**Table 1 biology-11-01438-t001:** International Shark Attack File: World unprovoked totals.

Year	Total Attacks	Fatal	Non-Fatal
2012	84	7	77
2013	77	10	67
2014	73	3	70
2015	8	6	92
2016	85	4	81
2017	90	5	85
2018	68	4	64
2019	67	2	65
2020	61	11	50
2021	73	9	64

## Data Availability

Data supporting reported results can be obtained for free by emailing the author at chris.pepin-neff@sydney.edu.au.

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
