# Peer review of "Shark Bite Reporting and The New York Times"

_biology, 2022, doi:10.3390/biology11101438_

Round 1

Reviewer 1 Report

I would like to congratulate the author on a very interesting study investigating an important, but underrepresented topic. It is very interesting to see how the terminology in the media is starting to change, allowing for a more diversified story telling. I also see the importance for scientists to support this change by incorporating and thus verifying descriptors for human-shark interactions that are less sensational and more accurately describing the nature of an interaction.

Overall, I very much appreciate the authors simplified approach, which makes the reported results very clear and obvious. However, one thing I would like to see in this approach if possible, would be the inclusion of a year prior prior to the introduction of the new descriptors in 2013 (e.g. 2012).

Furthermore, I think the author could improve upon the visual representation of the results. Namely Figure 1 and 2 could be made a bit more informative (for details on what I am referring to, please see the comments below). I would also like for the author to provide more references from the literature/media in the discussion and provide a bit more details on why this change in vocabulary is so important (for details, please see comments below).

Comments:

The following comments should be seen as suggestions.

Abstract:

Line 24 – 25 Consider starting a new sentence after 2021. It would make it easier to understand the second half of the current sentence. In its current state it doesn’t flow properly.

Introduction:

Line 46 “creature” instead of “create”

Line 59 “…into the reader’s subconscious…”

Line 73 Did you mean “…purpose…” instead of “…purchase…”?

Line 88 “…influence…” instead of “…influences…”

Line 96 The quotation marks are missing. Also, Are these categories from the 1958 symposium still in place? Whilst I have a relatively clear image of what a provoked and an unprovoked attack is, I would like to know more about how Sea and Air Disasters were defined. Maybe you could add another sentence here to clarify these two questions.

Line 140 I feel like there needs to be a better transition from the previous section to this one. It feels a bit jumpy. Maybe add a sentence prior to this line that introduces the following list of examples of institutes and the way they refer to shark incidents.

Line 141 Not sure what you mean with “…refer to shark bite/bite/bitten three times.”. I would simplify this and state that the Cape Cod National Parks Service refers to shark incidents as shark bites. Would be interesting to know how they refer to incidents that do not involve a shark biting a human.

Line 159 Please provide a timeframe for the data from the Australian Shark-Incident Database, i.e. years covered.

Line 159 Meaher, 2021 not in the reference list.

Line 165 – 166 I would rephrase this sentence a bit. For example: “Instead, when referring to human-shark interactions that resulted in injuries or death they used the terms “shark bite incidents” or “shark bites””

Line 171 – 172 This sentence could be moved up to where you described the changes in names of Shark-Incident Databases (Line 139), assuming that the International Shark Attack File in Florida not only maintains the verbiage, but also retains its name.

Materials and Methods

Line 197 Why are the material and methods reviewed? Unless required by the journal I would consider removing the first paragraph entirely.

Line 200 If the paragraph is being kept, “…engage…” instead of “…engages…”

Line 202 If the first paragraph is removed I would also suggest to remove “Identify the case study:”.

Line 217 – 218 Again, unless specifically asked for by the journal, I would remove “Operationalizing the data:” and the following sentence.

Line 219 “…looked at in context…”

Line 224 What do you mean with “Articles were removed to establish the five-year core articles.”? Does this mean you looked at all the articles from the New York Times between 2017 and 2021 that contained the phrase “shark attack”? If so, you should make that clearer. You should state that you sighted all articles from these years and you removed the once you described in lines 225 – 228 to establish your five-year core articles.

Results

Are you required to present the results as bullet points? If not I would much rather see this sections written in paragraphs. The issue with bullet points is that it is a list and we tend to lose proper sentence structure when writing it.

Line 235 Barnard, 2021 not in the reference list.

Line 236 “None of the articles were written by the same author twice.” – I am assuming that no article was written twice. Did you mean that there were no two articles that were written by the same author? In other words, each article was written by a different author.

Line 238 – 240 Could you present these findings in a graph? The fact that each article mentioning shark attacks also contained the term shark bite in the years 2018, 2020 and 2021 is very interesting and would be great to show in a figure.

Line 241 – 242 This is an interesting result, but in itself not very conclusive. How many shark incidents were there in 2018 compared to 2021? How was the ratio between the terms “shark attack” and “shark bite” in each year.

Line 243 Isn’t it the third most commonly used phrase? The most commonly used one still seems to be “shark attack”, based on Figure 2. Also, it only appears to be marginally more common compared to “Incident” and “Encounter”.

Line 246 I would not say that n=34 is on par with n=51.

Line 247 – 249 Out of how many articles? Could this be a bias related to the authors?

Figure 1 and Figure 2 First I would turn the order of the legends around. It feels off to start with 2021 on the left. Second, you need labels on your axes. Third, I would consider putting the years on the x-axis and use the colour coding for the different descriptors. That would provide an immediate overview of the ratios of descriptors within each year.

Line 258 So far you used the term human-shark interaction, does human-shark activity refer the same thing? If so, keep it consistent throughout to avoid confusion.

Line 259 – 260 What studies are you referring to here? Please provide references.

Line 260 – 262 You are talking about a trend in the media. Assuming this goes beyond the articles from the New York times you have been analysing I would like to see some references here.

Line 262 – 264 Why is a varied lexicon and vocabulary important? You could include the content of your last conclusion sentence here. I think it would fit much better and feel less attached. Try to synthesise this with the current literature and provide some references.

Line 264 – 266 Provide some references here to support your claim of scientific bodies focusing on the Associated Press Style Guide.

Line 267 – 268 If your data supports a layering approach to media reporting on human-shark interactions, you should provide some references for examples of such an approach being taken outside the New York times.

Line 270 Again, if activity is referring to the same thing as interaction, I would stick to one term to keep it consistent.

Line 272 Neff, 2016 not in the reference list.

Line 273 – 275 Is there a reference you could provide that would support this statement?

Line 275 – 277 First, you need to provide a reference for those 40% of reported shark “attacks” involving no injury. Second, I would state that if the public knew about this it could change a building block of marine science education. You cannot say with certainty that it would. Also, what do you mean with “…change a building block of marine science education.”? Maybe rephrase this a bit to make it clearer what you are trying to say.

Line 277 – 278 Again, provide some references here.

Line 292 – 294 This feels very much like an afterthought. I would move it up to the discussion (see comment on Line 262 – 264).

Author Response

September 22, 2022

RE: Re-submission of Brief Report
Word count: 5,191
Display items: 6 (5 figures/1 table

Please find the enclosed revised manuscript “Shark Bite Reporting and The New York Times” submitted as a Brief Report for review to the journal Biology for their Special Issue Understanding and Managing Human–Shark Interactions in an Environmentally Aware World.

I am grateful for the feedback from the reviewers which included 92 suggested edits to the manuscript. Revisions were made to nearly all suggestions and are outlined below as well as in the track-changes function of the main document text, which is attached.

 I am very pleased to report that based on the feedback from four reviewers, I have made major revisions to this manuscript for their consideration. This includes:

  1. 4 years of additional data from New York Times articles, totalling (2013-2021)
  2. 20 new citations to support the literature throughout the text; and
  3. 3 new figures to display the data.

In addition, I have updated the introduction and made a number of fixes to style and structure in keeping with recommendations from the reviewers.   

The new chart used in the text (Figure 3) illustrates the power of the new data which clearly illustrates an empirical change in NY Times media reporting to human-shark interactions after 2017.

I am grateful to Biology for the opportunity to revise this manuscript and look forward to the next step in this process. Again, I confirm that neither the manuscript nor any parts of its content are currently under consideration or published in another journal. All authors have approved the manuscript and agree with its submission to Biology.

Thank you for your consideration of this submission.

Sincerely,

Christopher Pepin-Neff, PhD
Senior Lecturer in Public Policy
Discipline of Government and International Relations
University of Sydney

Itemized responses to reviewer feedback:

REVIEWER 1

  1. Overall, I very much appreciate the authors simplified approach, which makes the reported results very clear and obvious.
  2. However, one thing I would like to see in this approach if possible, would be the inclusion of a year prior prior to the introduction of the new descriptors in 2013 (e.g. 2012). – Thank you. Five (5) years of media reporting data were added: 2012-2021.
  3. Furthermore, I think the author could improve upon the visual representation of the results. Namely Figure 1 and 2 could be made a bit more informative (for details on what I am referring to, please see the comments below).
  4. I would also like for the author to provide more references from the literature/media in the discussion and provide a bit more details on why this change in vocabulary is so important (for details, please see comments below). Thank you, Yes - 26 new references were added to this revised draft including many regarding media.

Comments:

Abstract:

  1. Line 24 – 25 Consider starting a new sentence after 2021. It would make it easier to understand the second half of the current sentence. In its current state it doesn’t flow properly. – Thank you. Yes - this confusing sentence has been deleted.

Introduction:

  1. Line 46 “creature” instead of “create” – Thank you. Changed
  2. Line 59 “…into the reader’s subconscious…” – Thank you. Changed
  3. Line 73 Did you mean “…purpose…” instead of “…purchase…”? -- Thank you, I have left this because purchase is the intended word in this case.
  4. Line 88 “…influence…” instead of “…influences…” -- Thank you. Changed
  5. Line 96 The quotation marks are missing. – Thank you. Changed
  6. Also, Are these categories from the 1958 symposium still in place? – Thank you. Yes, mostly. This text has been updated to specifically note the current use of unprovoked bites and provoked bites.
  7. Whilst I have a relatively clear image of what a provoked and an unprovoked attack is, I would like to know more about how Sea and Air Disasters were defined. Maybe you could add another sentence here to clarify these two questions. – Thank you. Please see above.
  8. Line 140 I feel like there needs to be a better transition from the previous section to this one. It feels a bit jumpy. Maybe add a sentence prior to this line that introduces the following list of examples of institutes and the way they refer to shark incidents. – Thank you. Yes – Changed.
  9. Line 141 Not sure what you mean with “…refer to shark bite/bite/bitten three times.”. I would simplify this and state that the Cape Cod National Parks Service refers to shark incidents as shark bites. – Thank you, Yes - Changed.
  10. Would be interesting to know how they refer to incidents that do not involve a shark biting a human. – Thank you. This was a hard one and there is no data available at present. I have requested the data from the International Shark Attack File.
  11. Line 159 Please provide a timeframe for the data from the Australian Shark-Incident Database, i.e. years covered. – Thank you, Yes – “1984” added.
  12. Line 159 Meaher, 2021 not in the reference list. -- Thank you, Added.
  13. Line 165 – 166 I would rephrase this sentence a bit. For example: “Instead, when referring to human-shark interactions that resulted in injuries or death they used the terms “shark bite incidents” or “shark bites”” Thank you. This is an editorial choice for the author that I would like to keep.
  14. Line 171 – 172 This sentence could be moved up to where you described the changes in names of Shark-Incident Databases (Line 139), assuming that the International Shark Attack File in Florida not only maintains the verbiage, but also retains its name. – Thank you, Yes - Changed

Materials and Methods

  1. Line 197 Why are the material and methods reviewed? Unless required by the journal I would consider removing the first paragraph entirely. – Thank you, this was kept per author guidelines.
  2. Line 200 If the paragraph is being kept, “…engage…” instead of “…engages…” – Thank you, Yes - changed.
  3. Line 202 If the first paragraph is removed I would also suggest to remove “Identify the case study:” – Thank you, please see above.
  4. Line 217 – 218 Again, unless specifically asked for by the journal, I would remove “Operationalizing the data:” and the following sentence. – Thank you, kept per author guidelines.
  5. Line 219 “…looked at in context…” – Thank you, Yes - Changed
  6. Line 224 What do you mean with “Articles were removed to establish the five-year core articles.”? Does this mean you looked at all the articles from the New York Times between 2017 and 2021 that contained the phrase “shark attack”? If so, you should make that clearer. You should state that you sighted all articles from these years and you removed the once you described in lines 225 – 228 to establish your five-year core articles. – Thank you, Yes - Revised and updated.

Results

  1. Are you required to present the results as bullet points? If not I would much rather see this sections written in paragraphs. The issue with bullet points is that it is a list and we tend to lose proper sentence structure when writing it. – Thank you, Kept in line with Author guidelines
  2. Line 235 Barnard, 2021 not in the reference list. Thank you, Added.
  3. Line 236 “None of the articles were written by the same author twice.” – I am assuming that no article was written twice. Did you mean that there were no two articles that were written by the same author? In other words, each article was written by a different author. – changed.
  4. Line 238 – 240 Could you present these findings in a graph? The fact that each article mentioning shark attacks also contained the term shark bite in the years 2018, 2020 and 2021 is very interesting and would be great to show in a figure. – Thank you, Yes – the Figures have been updated.
  5. Line 241 – 242 This is an interesting result, but in itself not very conclusive. How many shark incidents were there in 2018 compared to 2021? How was the ratio between the terms “shark attack” and “shark bite” in each year. Thank you -- Revised and added average year number of bites and fatalities
  6. Line 243 Isn’t it the third most commonly used phrase? The most commonly used one still seems to be “shark attack”, based on Figure 2. Also, it only appears to be marginally more common compared to “Incident” and “Encounter”.
  7. Line 246 I would not say that n=34 is on par with n=51. – Thank you. Yes - Changed.
  8. Line 247 – 249 Out of how many articles? Could this be a bias related to the authors? All are out of 14. -- Thank you. Additional articles and years were added to account for individual bias. In addition, no authors were the same for more than one article.
  9. Figure 1 and Figure 2 First I would turn the order of the legends around. It feels off to start with 2021 on the left. Second, you need labels on your axes. Third, I would consider putting the years on the x-axis and use the colour coding for the different descriptors. That would provide an immediate overview of the ratios of descriptors within each year. Thank you. Yes – the Figures have been updated.
  10. Line 258 So far you used the term human-shark interaction, does human-shark activity refer the same thing? If so, keep it consistent throughout to avoid confusion. – Thank you, Yes - Changed
  11. Line 259 – 260 What studies are you referring to here? Please provide references. – Thank you. Yes – additional citations have been added.
  12. Line 260 – 262 You are talking about a trend in the media. Assuming this goes beyond the articles from the New York times you have been analysing I would like to see some references here. Thank you. This is in reference to the changes in Australia, California, Cape Cod, and Cape Town noted above.
  13. Line 262 – 264 Why is a varied lexicon and vocabulary important? You could include the content of your last conclusion sentence here. I think it would fit much better and feel less attached. Try to synthesise this with the current literature and provide some references. Thank you. 26 additional references have been added to this draft revision including several around media literature on sharks.
  14. Line 264 – 266 Provide some references here to support your claim of scientific bodies focusing on the Associated Press Style Guide. Thank you, this is noted in the Shiffman quote.
  15. Line 267 – 268 If your data supports a layering approach to media reporting on human-shark interactions, you should provide some references for examples of such an approach being taken outside the New York times. Thank you. This is in reference to the changes in Australia, California, Cape Cod, and Cape Town noted above.
  16. Line 270 Again, if activity is referring to the same thing as interaction, I would stick to one term to keep it consistent. – Thank you, - Yes- Changed.
  17. Line 272 Neff, 2016 not in the reference list. Thank you, Yes – Added.
  18. Line 273 – 275 Is there a reference you could provide that would support this statement? [. ]
  19. Line 275 – 277 First, you need to provide a reference for those 40% of reported shark “attacks” involving no injury. – Thank you, Yes- added.
  20. Second, I would state that if the public knew about this it could change a building block of marine science education. You cannot say with certainty that it would. Also, what do you mean with “…change a building block of marine science education.”? Maybe rephrase this a bit to make it clearer what you are trying to say. – Thank you. Yes - Changed.
  21. Line 277 – 278 Again, provide some references here. Thank you, Yes – references have been added.
  22. Line 292 – 294 This feels very much like an afterthought. I would move it up to the discussion (see comment on Line 262 – 264). [ ]

REVIEWER #2:

The paper investigates an important aspect of shark conservation by addressing a large influential newspaper. There are some points that need to be considered to improve the understanding of the paper. At present, the writing veers between popular and scientific. For example, the language used is familiar in parts and scientific in others, and the third and first person reportage.

  1. General: the use of “data” as singular and plural is confusing throughout the manuscript

Introduction

  1. Line 39 needs referencing – Thank you, Yes – references added.
  2. Line 46: a create meaning a creature? – Thank you, Yes - changed
  3. Paragraph from Line 48: this seems more appropriate in the Discussion
  4. Line 70: would this be reference 3? – Thank you, Yes – References have been reworked.
  5. Line 112: reference needed -- Thank you, Yes – Added.
  6. Paragraph from Line 133: references needed rather than names only – Thank you, References have been added in keeping with Author Guidelines.
  7. Picture 1: far too small to be useful at present – Thank you, yes – an additional 5 years of data has been added.
  8. Line 185: Does this refer to the 2015 data or the previous data? Thank you, Yes – Exactly.
  9. Line 195: is this where reference to Picture 2 should be? I can’t find it in the text elsewhere. The caption needs more information to be useful. Thank you. Yes - Figures have been reworked.

Materials and Methods

  1. Line 198: the text seems to suggest that there are three sources of data, but the results only focus on the New York Times data. I am not clear what was being analysed. Thank you, Yes – this has been clarified.

Results

  1. I cannot see any reference to data from “Factiva and media reports” [Line 198]
    • Thank you, Yes - added this: New York Times articles were selected following a Factiva search that was refined based on the publication, publication date, and keywords.
  2. Line 232: who are “they”? Thank you. “They” refers to the descriptors in this case.
  3. Do the number of articles reflect the number of incidents or do some incidents result in more articles or a more keen reporter?
    • Thank you, Yes added this: The number of incidents per year on average are 74, which is greater than the number of New York Times articles on human-shark interactions each year.
  4. Line 279: the Introduction showed that the Australian situation has improved over the years and the Discussion appears to suggest that it is the New York Times that has not moved its position substantially. Yet the Discussion appears to suggest that there is an overall improvement.

Thank you, Added this: the data support an increase in shark bite usage at the New York Times. The New York Times, founded in 1851, as a reporting institution has 9 million subscribers and a high degree of continuity and a slow history of variation. The New York Times Manual of Style and Usage has been updated four times since 1950. As a result, these perceived adjustments and changes to human-shark interaction reporting are significant.

  1. The Neff & Hueter 2013 categories are very useful and could be used for other animal encounters, both water-based and terrestrial – Thank you, Yes -added.

REVIEWER 3:

Pepin-Neff provided a brief report focusing on the way the New York Times has been describing the human-shark interaction in the past 5 years. Although the manuscript has surely the potential to provide new important insights on the perceptions and way media are reporting events involving sharks, it needs an extensively revision before publication as at present is yet not at a standard sufficient for publication.

  1. As major points, I would suggest the author to re-organize the introduction section with a more proper and logical flow. For instance, the author could:
    1. Describe the background of this study (e.g. the use of phrase “shark attack” in the past and how this could influence the public perception – without any spoiler to results of the current study) that motivates the research question. – Thank you, revised and added. The thesis is kept in the body of the introduction.
    2. Describe why it is important to focus on the narratives used my media in the context of shark research and conservation. Thank you, Yes - added.
    3. Describe the aims of this study Thank you, Yes – added.
    4. Describe the importance of this study Thank you, Yes – added.
  2. I would also suggest the authors to avoid reporting quotes (see line 43-47; line 78-81; 83-86; 94-96) in the manuscript, especially in the introduction where the author reported at least 5 different statements. I think it would be much more explicative if the author takes the time to explain the background behind his study, and not simply reporting previous statements. I understand that this is a subject the author has been focusing on since long time, however the reader might not be so familiar and need to be guide with a logical and scientific flow. Thank you, Yes – point well taken. A lot of new text has been added to build around the quotes. But part of this is just style regarding the way one author builds a case. Nearly all other edits have been made.
  3. For instance (line 75-81) “…. following a highly publicized article in the Guardian and New York Times, co-median Stephen Colbert had a short segment on his show, “The Late Show” in July 2021 regarding attempts to relabel human-shark interactions, stating: “In order to change public perception of sharks, Australian scientists are seeking to rebrand shark attacks as interactions or incidents. As in, I’m sorry ma’am, a shark interacted with your husband’s torso and he’s experiencing a not being alive incident”. I think the author should re-write this sentence in a more scientific format. – Thank you, Please see above comment but I would just add that this joke and anecdote is essential to the point that people are making jokes at the expense of people working on this issue.

Material and methods: the authors should clarify the criteria used for classifying each article. What are these criteria that allow the author to distinguish between different type of discourse? See line 231-232. Thank you very much, Yes – this is a confusing sentence and has been edited to reflect that I am not looking at types of discourse, but rather usage of discourse.

Results and Discussion: the author need to carry out statistical analysis in support of the discussion section. – Thank you – this is an empirical qualitative analysis using a content analysis when is best suited to this type of review.

  1. Finally, the author should double check the Journal’s Instruction of Authors as some format and style needs to be modify (e.g. Figure 1 instead of Picture 1). Thank you – Yes – this has been changed.

Please find below other comments that might help the authors to improve the overall manuscript: 

  1. Line 13: I would suggest “This study found…” Thank you, Yes - Changed
  2. Line 41-43: “studies have shown…” which studies? the author needs to provide references in support to this statement. Thank you, Yes – references have been added.
  3. Line 43-47: I see the author often report quotes (e.g. line 78-81; 83-86; 94-96…) along the manuscript. I would avoid this. Thank you, noted and discussed above.
  4. Line 48-56: I would not anticipate any of the study’s results. This part should not be here. Introduction serves to the author to introduce the background of the focus of the study. Noted and discussed above.
  5. Line 57-59: similar to line 48-56, the author describes the importance of this study, however he hasn’t fully introduced the study’s focus yet. I would suggest to re-organize the introduction section such as suggested in the general comments. Thank you, Yes – the introduction was re-organized based on the outline the reviewer suggested.
  6. Line 155: I looked at the Journal’s guideline, and the author should use the term “Figure” instead of “Picture”, please modify it throughout the whole manuscript. Thank you, Yes - CHANGED.
  7. Line 174-177: I think the citation should be properly placed. Thank you, Yes - CHANGED
  8. Moreover, as for quotes, I would avoid question mark throughout the manuscript. Thank you – Yes, CHANGED
  9. I would keep a scientific language and form. Thank you, noted and important.
  10. Line 191: why choosing the New York Times? I think the author should provide some background in the introduction to explain why this choice and not another journal. Thank you, Yes - LANGUAGE ADDED.
  11. Line 197-201: this part is a repetition, please remove it. Thank you. CHANGED AND EDITED/REDUCED
  12. Line 202: “Identify the case study:” don’t need for this, the author should remove it. Same for “Gathering the data” and “Operationalizing the data:”Thank you. This section has been reduced but introductory markers have been left to note the three elements of case study examination.  
  13. Line 207: “A total of 14 New York Times newspaper articles…..” Thank you, Yes - CHANGED
  14. Line 215: “picture 2” please correct it Thank you, Yes - CHANGED
  15. Line 217-218: “Operationalizing the data: the data gathered answers the research question because these two chief phrases introduce the lexicon of human-shark interactions” don’t need for this line, the author should remove it. Thank you, Yes - CHANGED AND REDUCED/REVISED
  16. Line 230-231: don’t need for this, the author should remove it Thank you. Yes – Deleted.
  17. Line 231-233: “Data (descriptor mentions) were analyzed and separated for content analysis based on the way they highlighted different types of discourse.” The author should provide further details in the material and methods section. Thank you. Yes – additional revisions and updated to the methods section.
  18. Line 234-236: what about the other keywords?Thank you. Additional key words are noted further down.
  19. Line 237-240: in this case, is the author focusing on the headline or body text? Same for point 3-4 and 5? Thank you. Both the headline and body text is combined for each article.
  20. Line 235: please check the citation format Thank you, Yes – Checked w Author Guidelines.
  21. Line 251-253: I would suggest the author to combine figure 1 and 2 in one figure and create two panel: a and b[Thank you. This is useful feedback.]
  22. Results (general comments) – I would suggest the author to avoid using numbering in the result section and the author should implement this part with some statistics. Thank you, - this is in keeping with the Author Guidelines.
  23. Line 255-266: the author needs some statistical analyses in support of this statement. Thank you, Yes - STATEMENT REVISED.

Reviewer 4

This work is a subject that has been successfully expressed in terms of content and narration, and attracts the attention of the readers. In particular, it was necessary to investigate the negative reactions of the term "shark attack" on humans. The authors made an evaluation of the media reports published on this subject. When I checked the references in the article, I think that the reference stated in the results section (Barnard, 2021) in the 3rd article should be written clearly in the references section.

Thank you. All suggested edits were made.

Best Regards.

Reviewer 2 Report

The paper investigates an important aspect of shark conservation by addressing a large influential newspaper. There are some points that need to be considered to improve the understanding of the paper. At present, the writing veers between popular and scientific. For example, the language used is familiar in parts and scientific in others, and the third and first person reportage.

General: the use of “data” as singular and plural is confusing throughout the manuscript

Introduction

Line 39 needs referencing

Line 46: a create meaning a creature?

Paragraph from Line 48: this seems more appropriate in the Discussion

Line 70: would this be reference 3?

Line 112: reference needed

Paragraph from Line 133: references needed rather than names only

Picture 1: far too small to be useful at present

Line 185: Does this refer to the 2015 data or the previous data?

Line 195: is this where reference to Picture 2 should be? I can’t find it in the text elsewhere. The caption needs more information to be useful.

Materials and Methods

Line 198: the text seems to suggest that there are three sources of data, but the results only focus on the New York Times data. I am not clear what was being analysed.

Results

I cannot see any reference to data from “Factiva and media reports” [Line 198]

Line 232: who are “they”?

Do the number of articles reflect the number of incidents or do some incidents result in more articles or a more keen reporter?

Line 279: the Introduction showed that the Australian situation has improved over the years and the Discussion appears to suggest that it is the New York Times that has not moved its position substantially. Yet the Discussion appears to suggest that there is an overall improvement.

The Neff & Hueter 2013 categories are very useful and could be used for other animal encounters, both water-based and terrestrial.

Author Response

September 22, 2022

RE: Re-submission of Brief Report
Word count: 5,191
Display items: 6 (5 figures/1 table)

Please find the enclosed revised manuscript “Shark Bite Reporting and The New York Times” submitted as a Brief Report for review to the journal Biology for their Special Issue Understanding and Managing Human–Shark Interactions in an Environmentally Aware World.

I am grateful for the feedback from the reviewers which included 92 suggested edits to the manuscript. Revisions were made to nearly all suggestions and are outlined below as well as in the track-changes function of the main document text, which is attached.

 I am very pleased to report that based on the feedback from four reviewers, I have made major revisions to this manuscript for their consideration. This includes:

  1. 4 years of additional data from New York Times articles, totalling (2013-2021)
  2. 20 new citations to support the literature throughout the text; and
  3. 3 new figures to display the data.

In addition, I have updated the introduction and made a number of fixes to style and structure in keeping with recommendations from the reviewers.   

The new chart used in the text (Figure 3) illustrates the power of the new data which clearly illustrates an empirical change in NY Times media reporting to human-shark interactions after 2017.

I am grateful to Biology for the opportunity to revise this manuscript and look forward to the next step in this process. Again, I confirm that neither the manuscript nor any parts of its content are currently under consideration or published in another journal. All authors have approved the manuscript and agree with its submission to Biology.

Thank you for your consideration of this submission.

Sincerely,

Christopher Pepin-Neff, PhD
Senior Lecturer in Public Policy
Discipline of Government and International Relations
University of Sydney

Itemized responses to reviewer feedback:

REVIEWER 1

  1. Overall, I very much appreciate the authors simplified approach, which makes the reported results very clear and obvious.
  2. However, one thing I would like to see in this approach if possible, would be the inclusion of a year prior prior to the introduction of the new descriptors in 2013 (e.g. 2012). – Thank you. Five (5) years of media reporting data were added: 2012-2021.
  3. Furthermore, I think the author could improve upon the visual representation of the results. Namely Figure 1 and 2 could be made a bit more informative (for details on what I am referring to, please see the comments below).
  4. I would also like for the author to provide more references from the literature/media in the discussion and provide a bit more details on why this change in vocabulary is so important (for details, please see comments below). Thank you, Yes - 26 new references were added to this revised draft including many regarding media.

Comments:

Abstract:

  1. Line 24 – 25 Consider starting a new sentence after 2021. It would make it easier to understand the second half of the current sentence. In its current state it doesn’t flow properly. – Thank you. Yes - this confusing sentence has been deleted.

Introduction:

  1. Line 46 “creature” instead of “create” – Thank you. Changed
  2. Line 59 “…into the reader’s subconscious…” – Thank you. Changed
  3. Line 73 Did you mean “…purpose…” instead of “…purchase…”? -- Thank you, I have left this because purchase is the intended word in this case.
  4. Line 88 “…influence…” instead of “…influences…” -- Thank you. Changed
  5. Line 96 The quotation marks are missing. – Thank you. Changed
  6. Also, Are these categories from the 1958 symposium still in place? – Thank you. Yes, mostly. This text has been updated to specifically note the current use of unprovoked bites and provoked bites.
  7. Whilst I have a relatively clear image of what a provoked and an unprovoked attack is, I would like to know more about how Sea and Air Disasters were defined. Maybe you could add another sentence here to clarify these two questions. – Thank you. Please see above.
  8. Line 140 I feel like there needs to be a better transition from the previous section to this one. It feels a bit jumpy. Maybe add a sentence prior to this line that introduces the following list of examples of institutes and the way they refer to shark incidents. – Thank you. Yes – Changed.
  9. Line 141 Not sure what you mean with “…refer to shark bite/bite/bitten three times.”. I would simplify this and state that the Cape Cod National Parks Service refers to shark incidents as shark bites. – Thank you, Yes - Changed.
  10. Would be interesting to know how they refer to incidents that do not involve a shark biting a human. – Thank you. This was a hard one and there is no data available at present. I have requested the data from the International Shark Attack File.
  11. Line 159 Please provide a timeframe for the data from the Australian Shark-Incident Database, i.e. years covered. – Thank you, Yes – “1984” added.
  12. Line 159 Meaher, 2021 not in the reference list. -- Thank you, Added.
  13. Line 165 – 166 I would rephrase this sentence a bit. For example: “Instead, when referring to human-shark interactions that resulted in injuries or death they used the terms “shark bite incidents” or “shark bites”” Thank you. This is an editorial choice for the author that I would like to keep.
  14. Line 171 – 172 This sentence could be moved up to where you described the changes in names of Shark-Incident Databases (Line 139), assuming that the International Shark Attack File in Florida not only maintains the verbiage, but also retains its name. – Thank you, Yes - Changed

Materials and Methods

  1. Line 197 Why are the material and methods reviewed? Unless required by the journal I would consider removing the first paragraph entirely. – Thank you, this was kept per author guidelines.
  2. Line 200 If the paragraph is being kept, “…engage…” instead of “…engages…” – Thank you, Yes - changed.
  3. Line 202 If the first paragraph is removed I would also suggest to remove “Identify the case study:” – Thank you, please see above.
  4. Line 217 – 218 Again, unless specifically asked for by the journal, I would remove “Operationalizing the data:” and the following sentence. – Thank you, kept per author guidelines.
  5. Line 219 “…looked at in context…” – Thank you, Yes - Changed
  6. Line 224 What do you mean with “Articles were removed to establish the five-year core articles.”? Does this mean you looked at all the articles from the New York Times between 2017 and 2021 that contained the phrase “shark attack”? If so, you should make that clearer. You should state that you sighted all articles from these years and you removed the once you described in lines 225 – 228 to establish your five-year core articles. – Thank you, Yes - Revised and updated.

Results

  1. Are you required to present the results as bullet points? If not I would much rather see this sections written in paragraphs. The issue with bullet points is that it is a list and we tend to lose proper sentence structure when writing it. – Thank you, Kept in line with Author guidelines
  2. Line 235 Barnard, 2021 not in the reference list. Thank you, Added.
  3. Line 236 “None of the articles were written by the same author twice.” – I am assuming that no article was written twice. Did you mean that there were no two articles that were written by the same author? In other words, each article was written by a different author. – changed.
  4. Line 238 – 240 Could you present these findings in a graph? The fact that each article mentioning shark attacks also contained the term shark bite in the years 2018, 2020 and 2021 is very interesting and would be great to show in a figure. – Thank you, Yes – the Figures have been updated.
  5. Line 241 – 242 This is an interesting result, but in itself not very conclusive. How many shark incidents were there in 2018 compared to 2021? How was the ratio between the terms “shark attack” and “shark bite” in each year. Thank you -- Revised and added average year number of bites and fatalities
  6. Line 243 Isn’t it the third most commonly used phrase? The most commonly used one still seems to be “shark attack”, based on Figure 2. Also, it only appears to be marginally more common compared to “Incident” and “Encounter”.
  7. Line 246 I would not say that n=34 is on par with n=51. – Thank you. Yes - Changed.
  8. Line 247 – 249 Out of how many articles? Could this be a bias related to the authors? All are out of 14. -- Thank you. Additional articles and years were added to account for individual bias. In addition, no authors were the same for more than one article.
  9. Figure 1 and Figure 2 First I would turn the order of the legends around. It feels off to start with 2021 on the left. Second, you need labels on your axes. Third, I would consider putting the years on the x-axis and use the colour coding for the different descriptors. That would provide an immediate overview of the ratios of descriptors within each year. Thank you. Yes – the Figures have been updated.
  10. Line 258 So far you used the term human-shark interaction, does human-shark activity refer the same thing? If so, keep it consistent throughout to avoid confusion. – Thank you, Yes - Changed
  11. Line 259 – 260 What studies are you referring to here? Please provide references. – Thank you. Yes – additional citations have been added.
  12. Line 260 – 262 You are talking about a trend in the media. Assuming this goes beyond the articles from the New York times you have been analysing I would like to see some references here. Thank you. This is in reference to the changes in Australia, California, Cape Cod, and Cape Town noted above.
  13. Line 262 – 264 Why is a varied lexicon and vocabulary important? You could include the content of your last conclusion sentence here. I think it would fit much better and feel less attached. Try to synthesise this with the current literature and provide some references. Thank you. 26 additional references have been added to this draft revision including several around media literature on sharks.
  14. Line 264 – 266 Provide some references here to support your claim of scientific bodies focusing on the Associated Press Style Guide. Thank you, this is noted in the Shiffman quote.
  15. Line 267 – 268 If your data supports a layering approach to media reporting on human-shark interactions, you should provide some references for examples of such an approach being taken outside the New York times. Thank you. This is in reference to the changes in Australia, California, Cape Cod, and Cape Town noted above.
  16. Line 270 Again, if activity is referring to the same thing as interaction, I would stick to one term to keep it consistent. – Thank you, - Yes- Changed.
  17. Line 272 Neff, 2016 not in the reference list. Thank you, Yes – Added.
  18. Line 273 – 275 Is there a reference you could provide that would support this statement? [. ]
  19. Line 275 – 277 First, you need to provide a reference for those 40% of reported shark “attacks” involving no injury. – Thank you, Yes- added.
  20. Second, I would state that if the public knew about this it could change a building block of marine science education. You cannot say with certainty that it would. Also, what do you mean with “…change a building block of marine science education.”? Maybe rephrase this a bit to make it clearer what you are trying to say. – Thank you. Yes - Changed.
  21. Line 277 – 278 Again, provide some references here. Thank you, Yes – references have been added.
  22. Line 292 – 294 This feels very much like an afterthought. I would move it up to the discussion (see comment on Line 262 – 264). [ ]

REVIEWER #2:

The paper investigates an important aspect of shark conservation by addressing a large influential newspaper. There are some points that need to be considered to improve the understanding of the paper. At present, the writing veers between popular and scientific. For example, the language used is familiar in parts and scientific in others, and the third and first person reportage.

  1. General: the use of “data” as singular and plural is confusing throughout the manuscript

Introduction

  1. Line 39 needs referencing – Thank you, Yes – references added.
  2. Line 46: a create meaning a creature? – Thank you, Yes - changed
  3. Paragraph from Line 48: this seems more appropriate in the Discussion
  4. Line 70: would this be reference 3? – Thank you, Yes – References have been reworked.
  5. Line 112: reference needed -- Thank you, Yes – Added.
  6. Paragraph from Line 133: references needed rather than names only – Thank you, References have been added in keeping with Author Guidelines.
  7. Picture 1: far too small to be useful at present – Thank you, yes – an additional 5 years of data has been added.
  8. Line 185: Does this refer to the 2015 data or the previous data? Thank you, Yes – Exactly.
  9. Line 195: is this where reference to Picture 2 should be? I can’t find it in the text elsewhere. The caption needs more information to be useful. Thank you. Yes - Figures have been reworked.

Materials and Methods

  1. Line 198: the text seems to suggest that there are three sources of data, but the results only focus on the New York Times data. I am not clear what was being analysed. Thank you, Yes – this has been clarified.

Results

  1. I cannot see any reference to data from “Factiva and media reports” [Line 198]
    • Thank you, Yes - added this: New York Times articles were selected following a Factiva search that was refined based on the publication, publication date, and keywords.
  2. Line 232: who are “they”? Thank you. “They” refers to the descriptors in this case.
  3. Do the number of articles reflect the number of incidents or do some incidents result in more articles or a more keen reporter?
    • Thank you, Yes added this: The number of incidents per year on average are 74, which is greater than the number of New York Times articles on human-shark interactions each year.
  4. Line 279: the Introduction showed that the Australian situation has improved over the years and the Discussion appears to suggest that it is the New York Times that has not moved its position substantially. Yet the Discussion appears to suggest that there is an overall improvement.

Thank you, Added this: the data support an increase in shark bite usage at the New York Times. The New York Times, founded in 1851, as a reporting institution has 9 million subscribers and a high degree of continuity and a slow history of variation. The New York Times Manual of Style and Usage has been updated four times since 1950. As a result, these perceived adjustments and changes to human-shark interaction reporting are significant.

  1. The Neff & Hueter 2013 categories are very useful and could be used for other animal encounters, both water-based and terrestrial – Thank you, Yes -added.

REVIEWER 3:

Pepin-Neff provided a brief report focusing on the way the New York Times has been describing the human-shark interaction in the past 5 years. Although the manuscript has surely the potential to provide new important insights on the perceptions and way media are reporting events involving sharks, it needs an extensively revision before publication as at present is yet not at a standard sufficient for publication.

  1. As major points, I would suggest the author to re-organize the introduction section with a more proper and logical flow. For instance, the author could:
    1. Describe the background of this study (e.g. the use of phrase “shark attack” in the past and how this could influence the public perception – without any spoiler to results of the current study) that motivates the research question. – Thank you, revised and added. The thesis is kept in the body of the introduction.
    2. Describe why it is important to focus on the narratives used my media in the context of shark research and conservation. Thank you, Yes - added.
    3. Describe the aims of this study Thank you, Yes – added.
    4. Describe the importance of this study Thank you, Yes – added.
  2. I would also suggest the authors to avoid reporting quotes (see line 43-47; line 78-81; 83-86; 94-96) in the manuscript, especially in the introduction where the author reported at least 5 different statements. I think it would be much more explicative if the author takes the time to explain the background behind his study, and not simply reporting previous statements. I understand that this is a subject the author has been focusing on since long time, however the reader might not be so familiar and need to be guide with a logical and scientific flow. Thank you, Yes – point well taken. A lot of new text has been added to build around the quotes. But part of this is just style regarding the way one author builds a case. Nearly all other edits have been made.
  3. For instance (line 75-81) “…. following a highly publicized article in the Guardian and New York Times, co-median Stephen Colbert had a short segment on his show, “The Late Show” in July 2021 regarding attempts to relabel human-shark interactions, stating: “In order to change public perception of sharks, Australian scientists are seeking to rebrand shark attacks as interactions or incidents. As in, I’m sorry ma’am, a shark interacted with your husband’s torso and he’s experiencing a not being alive incident”. I think the author should re-write this sentence in a more scientific format. – Thank you, Please see above comment but I would just add that this joke and anecdote is essential to the point that people are making jokes at the expense of people working on this issue.

Material and methods: the authors should clarify the criteria used for classifying each article. What are these criteria that allow the author to distinguish between different type of discourse? See line 231-232. Thank you very much, Yes – this is a confusing sentence and has been edited to reflect that I am not looking at types of discourse, but rather usage of discourse.

Results and Discussion: the author need to carry out statistical analysis in support of the discussion section. – Thank you – this is an empirical qualitative analysis using a content analysis when is best suited to this type of review.

  1. Finally, the author should double check the Journal’s Instruction of Authors as some format and style needs to be modify (e.g. Figure 1 instead of Picture 1). Thank you – Yes – this has been changed.

Please find below other comments that might help the authors to improve the overall manuscript: 

  1. Line 13: I would suggest “This study found…” Thank you, Yes - Changed
  2. Line 41-43: “studies have shown…” which studies? the author needs to provide references in support to this statement. Thank you, Yes – references have been added.
  3. Line 43-47: I see the author often report quotes (e.g. line 78-81; 83-86; 94-96…) along the manuscript. I would avoid this. Thank you, noted and discussed above.
  4. Line 48-56: I would not anticipate any of the study’s results. This part should not be here. Introduction serves to the author to introduce the background of the focus of the study. Noted and discussed above.
  5. Line 57-59: similar to line 48-56, the author describes the importance of this study, however he hasn’t fully introduced the study’s focus yet. I would suggest to re-organize the introduction section such as suggested in the general comments. Thank you, Yes – the introduction was re-organized based on the outline the reviewer suggested.
  6. Line 155: I looked at the Journal’s guideline, and the author should use the term “Figure” instead of “Picture”, please modify it throughout the whole manuscript. Thank you, Yes - CHANGED.
  7. Line 174-177: I think the citation should be properly placed. Thank you, Yes - CHANGED
  8. Moreover, as for quotes, I would avoid question mark throughout the manuscript. Thank you – Yes, CHANGED
  9. I would keep a scientific language and form. Thank you, noted and important.
  10. Line 191: why choosing the New York Times? I think the author should provide some background in the introduction to explain why this choice and not another journal. Thank you, Yes - LANGUAGE ADDED.
  11. Line 197-201: this part is a repetition, please remove it. Thank you. CHANGED AND EDITED/REDUCED
  12. Line 202: “Identify the case study:” don’t need for this, the author should remove it. Same for “Gathering the data” and “Operationalizing the data:”Thank you. This section has been reduced but introductory markers have been left to note the three elements of case study examination.  
  13. Line 207: “A total of 14 New York Times newspaper articles…..” Thank you, Yes - CHANGED
  14. Line 215: “picture 2” please correct it Thank you, Yes - CHANGED
  15. Line 217-218: “Operationalizing the data: the data gathered answers the research question because these two chief phrases introduce the lexicon of human-shark interactions” don’t need for this line, the author should remove it. Thank you, Yes - CHANGED AND REDUCED/REVISED
  16. Line 230-231: don’t need for this, the author should remove it Thank you. Yes – Deleted.
  17. Line 231-233: “Data (descriptor mentions) were analyzed and separated for content analysis based on the way they highlighted different types of discourse.” The author should provide further details in the material and methods section. Thank you. Yes – additional revisions and updated to the methods section.
  18. Line 234-236: what about the other keywords?Thank you. Additional key words are noted further down.
  19. Line 237-240: in this case, is the author focusing on the headline or body text? Same for point 3-4 and 5? Thank you. Both the headline and body text is combined for each article.
  20. Line 235: please check the citation format Thank you, Yes – Checked w Author Guidelines.
  21. Line 251-253: I would suggest the author to combine figure 1 and 2 in one figure and create two panel: a and b[Thank you. This is useful feedback.]
  22. Results (general comments) – I would suggest the author to avoid using numbering in the result section and the author should implement this part with some statistics. Thank you, - this is in keeping with the Author Guidelines.
  23. Line 255-266: the author needs some statistical analyses in support of this statement. Thank you, Yes - STATEMENT REVISED.

Reviewer 4

This work is a subject that has been successfully expressed in terms of content and narration, and attracts the attention of the readers. In particular, it was necessary to investigate the negative reactions of the term "shark attack" on humans. The authors made an evaluation of the media reports published on this subject. When I checked the references in the article, I think that the reference stated in the results section (Barnard, 2021) in the 3rd article should be written clearly in the references section.

Thank you. All suggested edits were made.

Best Regards.

Reviewer 3 Report

General Comments:

Pepin-Neff provided a brief report focusing on the way the New York Times has been describing the human-shark interaction in the past 5 years. Although the manuscript has surely the potential to provide new important insights on the perceptions and way media are reporting events involving sharks, it needs an extensively revision before publication as at present is yet not at a standard sufficient for publication.

As major points, I would suggest the author to re-organize the introduction section with a more proper and logical flow. For instance, the author could:

a)    Describe the background of this study (e.g. the use of phrase “shark attack” in the past and how this could influence the public perception – without any spoiler to results of the current study) that motivates the research question

b)     Describe why it is important to focus on the narratives used my media in the context of shark research and conservation 

c)    Describe the aims of this study

d)    Describe the importance of this study

I would also suggest the authors to avoid reporting quotes (see line 43-47; line 78-81; 83-86; 94-96) in the manuscript, especially in the introduction where the author reported at least 5 different statements. I think it would be much more explicative if the author takes the time to explain the background behind his study, and not simply reporting previous statements. I understand that this is a subject the author has been focusing on since long time, however the reader might not be so familiar and need to be guide with a logical and scientific flow.

For instance (line 75-81) “…. following a highly publicized article in the Guardian and New York Times, co-median Stephen Colbert had a short segment on his show, “The Late Show” in July 2021 regarding attempts to relabel human-shark interactions, stating: “In order to change public perception of sharks, Australian scientists are seeking to rebrand shark attacks as interactions or incidents. As in, I’m sorry ma’am, a shark interacted with your husband’s torso and he’s experiencing a not being alive incident”. I think the author should re-write this sentence in a more scientific format.

Material and methods: the authors should clarify the criteria used for classifying each article. What are these criteria that allow the author to distinguish between different type of discourse? See line 231-232.

Results and Discussion: the author need to carry out statistical analysis in support of the discussion section.

Finally, the author should double check the Journal’s Instruction of Authors as some format and style needs to be modify (e.g. Figure 1 instead of Picture 1). 

Please find below other comments that might help the authors to improve the overall manuscript: 

Line 13: I would suggest “This study found…”

Line 41-43: “studies have shown…” which studies? the author needs to provide references in support to this statement

Line 43-47: I see the author often report quotes (e.g. line 78-81; 83-86; 94-96…) along the manuscript. I would avoid this.

Line 48-56: I would not anticipate any of the study’s results. This part should not be here. Introduction serves to the author to introduce the background of the focus of the study.

Line 57-59: similar to line 48-56, the author describes the importance of this study, however he hasn’t fully introduced the study’s focus yet. I would suggest to re-organize the introduction section such as suggested in the general comments.

Line 155: I looked at the Journal’s guideline, and the author should use the term “Figure” instead of “Picture”, please modify it throughout the whole manuscript.

Line 174-177: I think the citation should be properly placed. Moreover, as for quotes, I would avoid question mark throughout the manuscript. I would keep a scientific language and form. 

Line 191: why choosing the New York Times? I think the author should provide some background in the introduction to explain why this choice and not another journal.

Line 197-201: this part is a repetition, please remove it.

Line 202: “Identify the case study:” don’t need for this, the author should remove it. Same for “Gathering the data” and “Operationalizing the data:” 

Line 207: “A total of 14 New York Times newspaper articles…..”

Line 215: “picture 2” please correct it

Line 217-218: “Operationalizing the data: the data gathered answers the research question because these two chief phrases introduce the lexicon of human-shark interactions” don’t need for this line, the author should remove it.

Line 230-231: don’t need for this, the author should remove it

Line 231-233: “Data (descriptor mentions) were analyzed and separated for content analysis based on the way they highlighted different types of discourse.” The author should provide further details in the material and methods section.

Line 234-236: what about the other keywords? 

Line 237-240: in this case, is the author focusing on the headline or body text? Same for point 3-4 and 5?

Line 235: please check the citation format

Line 251-253: I would suggest the author to combine figure 1 and 2 in one figure and create two panel: a and b 

Results (general comments) – I would suggest the author to avoid using numbering in the result section and the author should implement this part with some statistics.

Line 255-266: the author needs some statistical analyses in support of this statement.

Author Response

(The authors gave the same response as above.)

Reviewer 4 Report

Dear Author/Authors,

This work is a subject that has been successfully expressed in terms of content and narration, and attracts the attention of the readers. In particular, it was necessary to investigate the negative reactions of the term "shark attack" on humans. The authors made an evaluation of the media reports published on this subject. When I checked the references in the article, I think that the reference stated in the results section (Barnard, 2021) in the 3rd article should be written clearly in the references section.

Best Regards.

Author Response

(The authors gave the same response as above.)

Round 2

Reviewer 3 Report

The author carefully addressed all my raised points and provided a satisfying revision.